# Complex Component of Oncocytic and Non-Oncocytic Lipoadenomas in the Parotid Gland: A Case Report

**DOI:** 10.3390/diagnostics11081478

**Published:** 2021-08-15

**Authors:** Fuyuki Sato, Takashi Nakajima, Takashi Sugino

**Affiliations:** Department of Diagnostic Pathology, Shizuoka Cancer Center, 1007 Shimonagakubo, Nagaizumi-cho, Sunto-gun, Shizuoka Prefecture 411-8777, Japan; t.nakajima@scchr.jp (T.N.); t.sugino@scchr.jp (T.S.)

**Keywords:** oncocytic lipoadenoma, salivary gland tumor, immunohistochemistry

## Abstract

Oncocytic lipoadenoma of the salivary gland is a rare tumor that develops mainly in the parotid gland. We report a case of oncocytic lipoadenoma of the parotid gland in a 70-year-old woman. The tumor measured 30 × 20 mm and had a well-circumscribed tan-brown surface. The tumor was histologically composed of oncocytic and lipomatous lesions without atypia. In addition to the oncocytic lipoadenoma, a small lipomatous tumor, measuring 10 × 7 mm, was found in the resected parotid gland. Macroscopically, this tumor was yellow and indistinguishable from the parotid gland. Microscopically, the tumor was rich in fats and contained an area of conglomerated duct-like proliferation and salivary gland components. Therefore, the tumor was diagnosed as a non-oncocytic lipoadenoma with a sialoadenoma component. We report the first case of double component oncocytic and non-oncocytic lipoadenomas of the salivary gland.

## 1. Introduction

Oncocytic lipoadenoma of the salivary glands is rare. Hirokawa et al. first reported oncocytic lipoadenoma of the submandibular gland as an uncommon salivary gland tumor [1]. Less than 30 cases have been reported in the literature. Most oncocytic lipoadenomas have occurred in the parotid gland. They are benign tumors mainly composed of oncocytes with various fatty components [2,3,4,5,6]. We encountered a case of oncocytic lipoadenoma, whose histologic findings resembled salivary oncocytoma. A non-oncocytic lipomatous salivary gland tumor was found adjacent to the oncocytic lipoadenoma. Non-oncocytic salivary tumors are predominantly composed of lipomatous component without oncocytes [7]. Lipomatous salivary gland tumors are rare lesions that sometimes consist of salivary gland elements, such as acini and ductal cells [7]. To the best of our knowledge, double component oncocytic and non-oncocytic lipomatous salivary gland tumors have not previously been reported. We report the first case of complex component oncocytic and lipomatous salivary gland tumors in the parotid gland.

## 2. Clinical Summary

A 70-year-old woman noticed a slow-growing, painless mass in her left cheek two years before hospital admission. Magnetic resonance imaging (MRI) revealed a 30 × 20-mm tumor in the superficial lobe of the left parotid gland. Gadolinium-enhanced T1-weighted imaging revealed a well-circumscribed tumor (Figure 1A-a). T2-weighted imaging documented an aberrant salivary duct (Figure 1A-b). The components inside the tumor were heterogeneous. The MRI findings suggested a hamartoma and benign tumor. A parotitis was excluded because it was a mass and well-circumscribed. A superficial parotidectomy was performed.

## 3. Pathological Findings

The tumor was macroscopically resected with the superficial lobe of the parotid gland. The tumor was a well-circumscribed, homogeneous, tan-brown, solid mass on the cut surface (Figure 1B). Microscopically, the tumor was surrounded by a thin fibrous capsule and composed of an admixed population of oncocytes and adipocytes in varying proportions (Figure 2A-a,b). The oncocytes were round to polygonal, had abundant granular eosinophilic cytoplasm, and did not exhibit cellular atypia. The oncocytes were arranged in small, compact, acinar and trabecular-solid patterns. The adipocytes resembled mature fat tissue. The proportion of oncocytes and adipocytes varied (Figure 2A-c). In the large area of the tumor, the oncocytes were more predominant than the adipocytes (Figure 2A-a). On immunohistochemistry, the cytokeratin 5/6 (CK5/6) and p63 stains were positive around the nests or acini of the oncocytes (Figure 2d, IHC). As shown in Figure 2e,f small amounts of serous acini and ducts of the salivary gland were observed, but sebaceous duct metaplasia and lymphocyte aggregates were absent. Alpha-smooth muscle actin (ASMA), discovered on Gastrointestinal stromal tumor-1 (DOG1) and S100, were negative in oncocytes (Figure 3d). We performed immunohistochemistry using an auto-stainer with automated protocols and the information was summarized in Table 1.

There was another circumscribed mass, measuring 10 × 7 mm, in the parotid gland tissue near the oncocytic tumor. It had a yellowish cut surface indistinguishable from the normal parotid gland tissue (Figure 1B). Microscopically, the tumor was separated into thin fibrous capsules from the parotid gland tissue and composed of large amounts of adipose tissue and small amounts of scattered salivary gland components, such as ducts and serous acini. This was identified as a non-oncocytic tumor. The non-oncocytic tumor was not continuous with the oncocytic tumor. The two tumors were separated by about 0.6 mm. As seen in Figure 2b,f, there was an island-like large fibrotic area containing two foci of conglomerated duct-like proliferation. Two foci of conglomerated duct-like proliferation were connected with irregularly branched salivary duct-like structures in the fibrotic background with lymphocyte aggregation. The duct-like structure was composed of double-layered cells. The nuclei of the cells were similar to normal duct cells, but their nuclear-cytoplasmic ratios were slightly higher. The peripheral cells of the conglomerated duct-like proliferation were immunohistochemically positive for p63, CK5/6 and ASMA. but negative for DOG1 (Figure 2f, IHC and Figure 3f). S100 was focally positive in inner cells. There were no oncocytic components. We summarized immunohistochemical data in oncocytic and non-oncocytic components (Table 2).

## 4. Discussion

In 1998, Hirokawa et al. first reported oncocytic lipoadenoma of the submandibular gland as a unique benign tumor composed of lipomatous and epithelial elements arising from the submandibular gland [1]. In 2001, Nagao et al. proposed a new category of sialolipoma, a distinct variant of salivary gland lipoma with secondary entrapment of salivary gland elements [8]. These two related salivary gland tumors, sialoadenoma and oncocytic lipoadenoma, were classified under sialolipoma. In 2017, World Health Organization classification of head and neck tumors described the lipoma/sialolipoma category as a benign soft tissue lesion of the salivary gland. Therefore, neoplastic lipomatous growth within major salivary glands, such as salivary lipoma, adenolipoma, oncocytic sialolipoma and oncocytic lipoadenoma, were classified as lipomas/sialolipomas.

Lipomatous salivary gland tumors are uncommon. Agaimy et al. categorized lipomatous salivary gland tumors into three main groups: ordinary lipoma, oncocytic lipoadenoma and non-oncocytic sialolipoma, based on the proportion and distribution of adipose tissue and epithelial type [7]. Oncocytic lipoadenoma was reportedly an epithelial-predominant tumor and was distinct from fat-dominated non-oncocytic sialolipoma. Lau et al. clinicopathologically analyzed seven cases of oncocytic lipoadenoma of the salivary gland. They concluded that oncocytic lipoadenoma had a distinctive morphologic appearance and should be separated from other salivary gland neoplasms with a prominent oncocytic component [2].

The oncocytic tumor in the present case was morphologically compatible with oncocytic lipoadenoma, as reported in a previous study [6,9]. Tan-brown macroscopic appearance and the microscopically predominant oncocyte-rich area suggested a salivary gland oncocytoma. The immunohistochemical results of CK5/6, p63, ASMA and DOG1 in this tumor confirmed that it was an oncocytoma. CK5/6 mainly stained with the basal cells of the neoplastic acini, while p63 stained with the nuclear oncocytic cells [10]. ASMA and DOG1 were negative for oncocytoma [11,12]. However, the presence of adipocytes and normal salivary gland tissues in the tumor were not characteristic of a salivary gland oncocytoma. It has been reported that oncocytic metaplasia associated with repeated oxidative stress during cellular aging [13]. Therefore, oxidative stress may affect the process of oncocytic lipoadenoma.

In addition to oncocytic lipoadenoma, there was another small lipomatous tumor in the resected parotid gland. This tumor was predominantly composed of adipocytes containing salivary gland components without oncocytes. Its morphology was identical to that of sialolipomas. This lipomatous tumor had a focus of conglomerated duct-like proliferation, forming a “nodule in nodule” appearance; in other words, a nodule of duct-like proliferation surrounded by a larger nodule of lipomatous component. The cells of the conglomerated duct-like proliferation were immunohistochemically positive for ASMA and S100, suggesting a myoepithelial and salivary gland nature. It has been reported S100 was positive by immunohistochemistry in many salivary gland tumors [14,15]. These results suggested that conglomerated duct-like proliferation was a neoplasm or adenoma. Therefore, this tumor was identified as a non-oncocytic lipoadenoma with a sialoadenoma component.

Differential diagnosis is parotitis and sialoangiolipoma. Parotitis shows inflammatory cells surrounding ducts in the parotid gland and no tumor components, such as oncocyte and abnormal adipose tissue. Sialoangiolipoma is a rare mesenchymal tumor resembling the non-oncocytic component of our case [16]. Although it includes acini and ducts, they were surrounded by variably sized blood vessels.

Here, we have reported unique complex components of rare oncocytic and non-oncocytic lipoadenomas of the salivary gland.

## Figures and Tables

**Figure 1 diagnostics-11-01478-f001:**
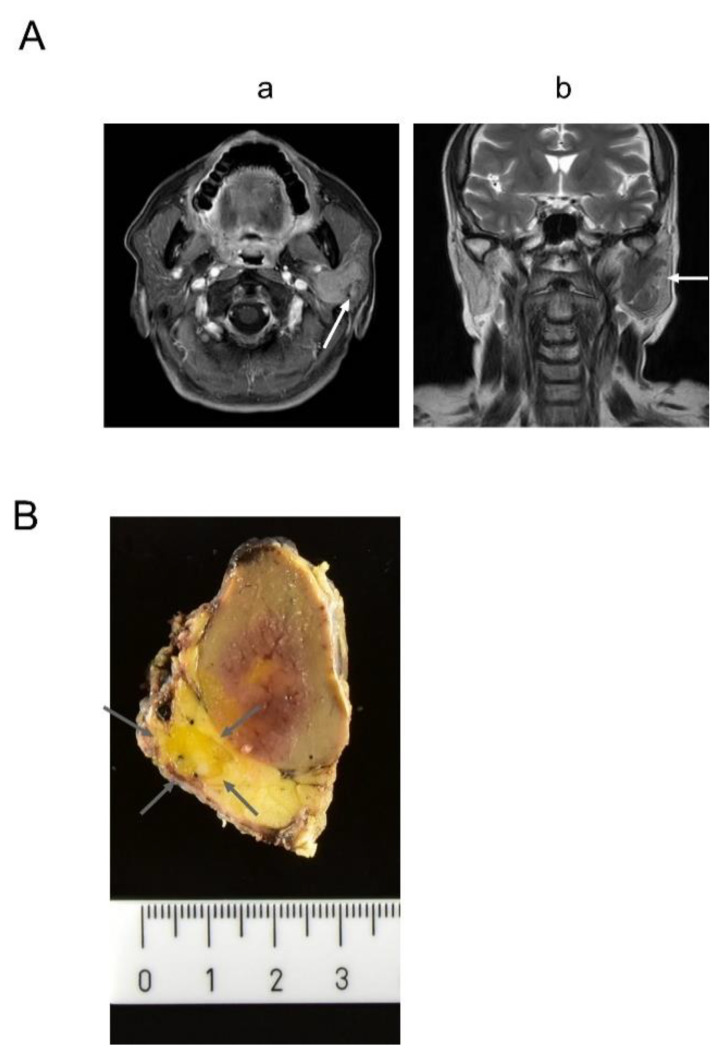
(**A**) Radiological images. (**a**) Gadolinium-enhanced T1-weighted imaging. The arrow shows a well-circumscribed tumor in the left parotid gland. (**b**) T2-weighted imaging. The arrow suggests an aberrant salivary duct. (**B**) Macroscopic images. A well-circumscribed brownish tumor with hemorrhage was seen adjacent to the parotid gland. The arrow shows a clouding yellow cut surface within the parotid gland.

**Figure 2 diagnostics-11-01478-f002:**
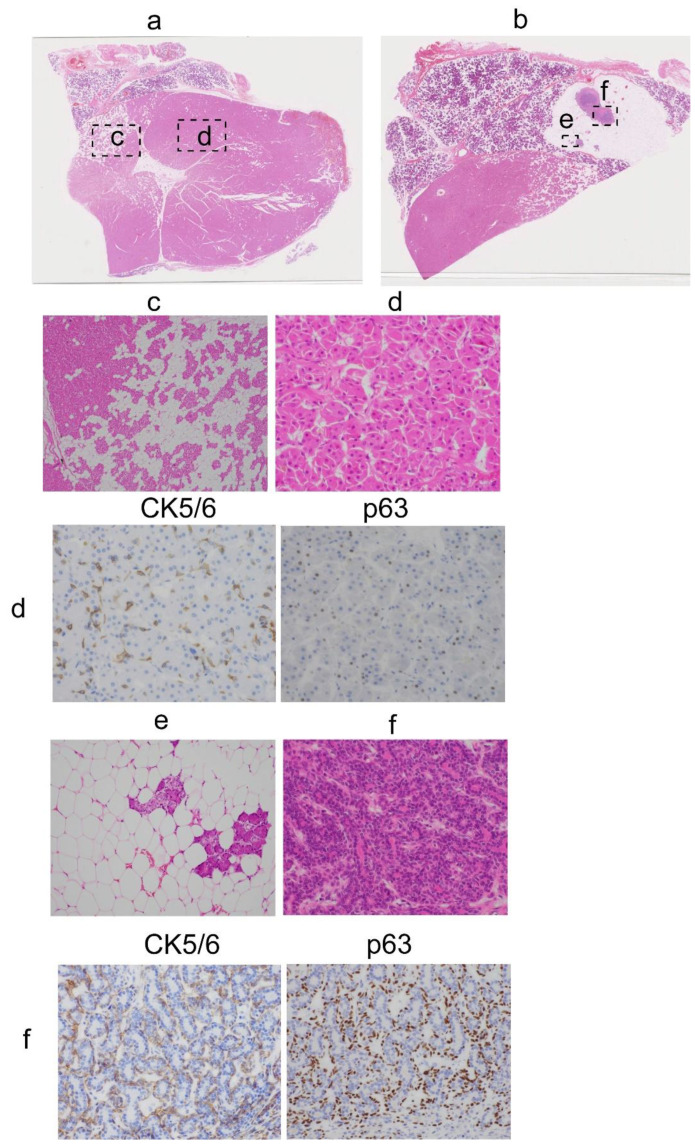
Histological aspects of oncocytic lipoadenoma. (**a**,**b**) Representative HE images were taken by virtual slide. Inset dot boxes were taken by microscope as below. (**c**) Mixed oncocytic and lipomatous components of the tumor. 40× magnification. (**d**) Large magnification of oncocytes. Immunoreactivities of CK5/6 and p63 in the oncocytic component. 400× magnification. (**e**) Acinar glands in non-oncocytic lipoadenoma. (**f**) Proliferated ductal cells in non-oncocytic lipoadenoma. Immunoreactivities of CK5/6 and p63 in ducts “nodule-in-nodule“ structure. 400× magnification.

**Figure 3 diagnostics-11-01478-f003:**
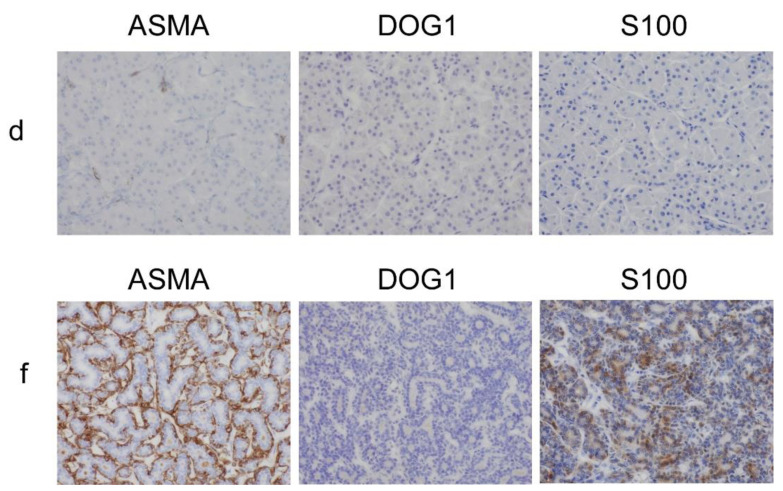
(**d**) Immunoreactivities of ASMA, DOG1 and S100 in the oncocytic component. 400× magnification. (**f**) Immunoreactivities of ASMA, DOG1 and S100 in ducts “nodule-in-nodule“ structure. 400× magnification.

**Table 1 diagnostics-11-01478-t001:** The summary of antigen retrieval. PA primary antibody, ER1 Epitope. retrieval 1 pH6.0, ER2 Epitope retrieval 2 pH9.0, CB Citrate buffer pH 6.0. Antigen retrieval of CK5/6, p63, ASMA and DOG1 was performed by autoclave. The fixation was performed with 10% neutral buffered formalin.

Antibody	CK5/6	p63	ASMA	DOG1	S100
Source	DAKO (M7237)	DAKO (M7317)	DAKO (M0851)	Nichirei (418041)	DAKO (IR504)
PA dilution	1:400	1:300	1:100	1:2	Ready to use
Autostainer	Leica (BOND-III)	Leica (BOND-III)	Leica (BOND-III)	Nichirei (Histostainer)	Leica (BOND-III)
Antigen retrieval	ER1 (30 min)	ER2 (20 min)	CB (10 min)	ER2 (10 min)	No antigen retrieval

**Table 2 diagnostics-11-01478-t002:** Summary of immunohistochemical data in oncocytic and non-oncocytic components.

	Lesion	Oncocytic	Non-oncocytic
Antibody			
CK5/6		Positive	Positive
p63		Positive	Positive
ASMA		Negative	Positive
DOG1		Negative	Negative
S100		Negative	Positive

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
