# Peer review of "Complex Component of Oncocytic and Non-Oncocytic Lipoadenomas in the Parotid Gland: A Case Report"

_diagnostics, 2021, doi:10.3390/diagnostics11081478_

Round 1

Reviewer 1 Report

Sir, 

I have recently reviewed the manuscript of a case report  "Simultaneous development of oncocytic and non-oncocytic lipoadenomas in the parotid gland: A case report" submitted by Fuyuki Sato and coworkers to Diagnostics. 

The authors present a single case of a rare neoplastic disorder affecting the parotid gland in a 70-year-old woman.  It is presented as a simultaneous development of oncocytic and non-oncocytic lipomatous salivary gland tumours. The authors believe that this feature has not been reported previously. 

The case is presented concisely. And it is, indeed, an interesting communication. These anecdotal publications of rare cases might be a valuable contribution to medical knowledge one day.

However, I believe that this story should be improved and further perfected. 

a) in their differential diagnosis, the authors also speculate about parotitis. For the sake of beginners, this should be excluded before the intervention (superficial parotidectomy). Please,  elaborate on this aspect. 

b) the immunohistochemical profile is relatively narrow. K5/6, p63. DOG1 and ASMA (n.b. the abbreviation must be explained after the very first use).  I miss at least S100 protein staining - S100 protein was proposed to assist the subclassification of salivary gland neoplasms with oncocytic differentiation. Why is DOG1 not presented?  And also ASMA? 

I believe that a comparative marker panel would be greatly helpful to novice readers. The salivary tumours immunohistochemical profile (in a table form) would be, therefore, a significant improvement. Generally, authors should suggest an optimal IHC panel to assess these rare lesions in salivary glands and discuss the purpose of each marker. 

c) IHC is an ancillary method  - anyway, it is not described (at least summary table of antibodies, dilutions, fixation and retrieval method etc.).  This is an indispensable part of any manuscript presenting IHC!  

d)  Oncocytic metaplasia represents a histopathologic feature that can be observed in normal tissue such as salivary and lacrimal glands. However, this sort of metaplasia may also constitute a degenerative process. This is recently understood as a result of repeated oxidative damage during cellular ageing. The authors should clearly discuss this issue and recent view on the nature of oncocytic tumours - and thus increase the educative quality of their manuscript. 

e) The authors conclude that they presented, and I quote, "simultaneous development of rare oncocytic and non-oncocytic lipoadenomas".  After all, I am not convinced about this aspect; this conclusion is not properly evidenced.   Authors correctly comment on the morphology of a “nodule in nodule”. It highlights the complex morphology of that particular case - right. But authors do not present any evidence of the temporal or molecular independent yet synchronous onset.  As it is, the conclusion seems to be overambitious. Therefore I would recommend reconsidering this aspect (and also headline accordingly). 

I believe that this manuscript requires some changes, as suggested above. Once the authors finish their work, I am happy to see it again.

Author Response

a) in their differential diagnosis, the authors also speculate about parotitis. For the sake of beginners, this should be excluded before the intervention (superficial parotidectomy). Please,  elaborate on this aspect. 

Response: Thank you for pointing this out. We described the parotitis in Clinical summary and Discussion.

“A parotitis was excluded because it was mass and well- circumscribed.”

“Differential diagnosis is parotitis and sialoangiolipoma. Parotitis shows inflammatory cells surrounding ducts in parotid gland and no tumor components such as oncocyte and abnormal adipose tissue.”

b) the immunohistochemical profile is relatively narrow. K5/6, p63. DOG1 and ASMA (n.b. the abbreviation must be explained after the very first use).  I miss at least S100 protein staining - S100 protein was proposed to assist the subclassification of salivary gland neoplasms with oncocytic differentiation. Why is DOG1 not presented?  And also ASMA? 

I believe that a comparative marker panel would be greatly helpful to novice readers. The salivary tumours immunohistochemical profile (in a table form) would be, therefore, a significant improvement. Generally, authors should suggest an optimal IHC panel to assess these rare lesions in salivary glands and discuss the purpose of each marker. 

Response: Thank you for pointing this out. We added the immunohistochemical data of ASMA, DOG1 and S100 (Fig. 3). We also made IHC marker panel in table 2.

c) IHC is an ancillary method  - anyway, it is not described (at least summary table of antibodies, dilutions, fixation and retrieval method etc.).  This is an indispensable part of any manuscript presenting IHC!  

Response: According to reviewer’s suggestion, we made summarized antibody condition in table1. We performed IHC using auto-stainer.

d)  Oncocytic metaplasia represents a histopathologic feature that can be observed in normal tissue such as salivary and lacrimal glands. However, this sort of metaplasia may also constitute a degenerative process. This is recently understood as a result of repeated oxidative damage during cellular ageing. The authors should clearly discuss this issue and recent view on the nature of oncocytic tumours - and thus increase the educative quality of their manuscript. 

Response: Thank you for raising this interesting point. We added the sentences in Discussion.

“It has been reported oncocytic metaplasia associated with repeated oxidative stress during cellular aging [13]. Therefore, oxidative stress may affect the process of oncocytic lipoadenoma.”

e) The authors conclude that they presented, and I quote, "simultaneous development of rare oncocytic and non-oncocytic lipoadenomas".  After all, I am not convinced about this aspect; this conclusion is not properly evidenced.   Authors correctly comment on the morphology of a “nodule in nodule”. It highlights the complex morphology of that particular case - right. But authors do not present any evidence of the temporal or molecular independent yet synchronous onset.  As it is, the conclusion seems to be overambitious. Therefore I would recommend reconsidering this aspect (and also headline accordingly). 

Response: Thank you for your suggestion. We corrected the conclusion in Discussion.

 “We report here unique complex component of rare oncocytic and non-oncocytic lipoadenomas of the salivary gland”.

 We also explained the sentences “Nodule in nodule” in Discussion.

“In other words, nodule of duct-like proliferation surrounded by larger nodule of lipomatous component”.

Thank you for reading our manuscript well.

Reviewer 2 Report

Dear Authors

thank you for your paper. I've found it very interesting and very well presented. I can suggest to evaluate this paper and if you mean interesting to discuss further among different salivary lipomatous tumours (Maiorano E, Capodiferro S, Fanelli B, Calabrese L, Napoli A, Favia G. Hamartomatous angiolipoma of the parotid gland (sialoangiolipoma), and if possible to improve quality of the histological pictures.

Thank you again for your paper  

Author Response

thank you for your paper. I've found it very interesting and very well presented. I can suggest to evaluate this paper and if you mean interesting to discuss further among different salivary lipomatous tumours (Maiorano E, Capodiferro S, Fanelli B, Calabrese L, Napoli A, Favia G. Hamartomatous angiolipoma of the parotid gland (sialoangiolipoma), and if possible to improve quality of the histological pictures.

Thank you again for your paper  

Response: Thank you for reading our manuscript well. We made more good quality of histological pictures. Also, we discussed the differences in sialoangiolipoma in Discussion.

“Differential diagnosis is parotitis and sialoangiolipoma. Parotitis shows inflammatory cells surrounding ducts in parotid gland and no tumor components such as oncocyte and abnormal adipose tissue. Sialoangiolipoma is rare mesenchymal tumor resembling with non-oncocytic component of our case [16]. Although it includes acini and ducts, they were surrounded by variably sized blood vessels”.

Thank you for your kindness.

Round 2

Reviewer 1 Report

Sir,

I have recently reviewed the revised version of the manuscript "diagnostics-1327377" - recently titled: "Complex component of oncocytic and non-oncocytic lipoadenomas in the parotid gland: A case report"  submitted by Fuyuki Sato, Takashi Nakajima and Takashi Sugino to Diagnostics. 

I have studied their rebuttal letter and I believe the authors dealt honestly with my previous critical points and suggestions. I am grateful for this modification and I believe it helped to improve the manuscript. 

The authors prepared Table 1, however, I believe this is not perfect. I believe, the epitope retrieval was performed in an autoclave (for all indicated antibodies where necessary), the variability comes with pH of the buffer. ER2 is not presented in the table but is mentioned in the table description. This requires clarification. Also, I believe it would be more accurate to use 10%NBF (neutral buffered formalin) instead of a somewhat colloquial 10% formalin (please verify if this was really used). For S100 protein, it would be more accurate to use "ready to use" instead of "no dilution". The first sentence of the legend is probably incomplete (missing a verb). The automated stainer type (line 4) seems to be less important information (and could be easily omited from the table). 

In general, once these petit changes are done, this interesting case report could be considered for publication. 

Author Response

The authors prepared Table 1, however, I believe this is not perfect. I believe, the epitope retrieval was performed in an autoclave (for all indicated antibodies where necessary), the variability comes with pH of the buffer. ER2 is not presented in the table but is mentioned in the table description. This requires clarification. Also, I believe it would be more accurate to use 10%NBF (neutral buffered formalin) instead of a somewhat colloquial 10% formalin (please verify if this was really used). For S100 protein, it would be more accurate to use "ready to use" instead of "no dilution". The first sentence of the legend is probably incomplete (missing a verb). The automated stainer type (line 4) seems to be less important information (and could be easily omited from the table). 

In general, once these petit changes are done, this interesting case report could be considered for publication. 

Response: Thank you for your suggestion. We corrected table1 and the legend as below.

“Table 1. The summary of antigen retrieval. PA primary antibody, ER1

Epitope retrieval 1 pH6.0, ER2 Epitope retrieval 2 pH9.0, CB Citrate buffer pH 6.0. Antigen retrieval of CK5/6, p63, ASMA and DOG1 was performed by autoclave. The fixation was performed with 10% neutral buffered formalin.”
